# Evaluation of emerging inflammatory markers for predicting oxygen support requirement in COVID-19 patients

Peerapong Kamjai[1,2,3], Sivaporn Hemvimol[4], Narisa Kengtrong Bordeerat[1], Potjanee Srimanote[5,6], Pornpimon Angkasekwinai[1,2,6]*

1 Department of Medical Technology, Faculty of Allied Health Sciences, Thammasat University, Pathumthani, Thailand, 2 Graduate Program in Medical Technology, Faculty of Allied Health Sciences, Thammasat University, Pathumthani, Thailand, 3 Department of Medical Technology and Clinical Pathology, Saraburi Hospital, Saraburi, Thailand, 4 Department of Medicine, Saraburi Hospital, Saraburi, Thailand, 5 Graduate Program in Biomedical Sciences, Faculty of Allied Health Sciences, Thammasat University, Pathumthani, Thailand, 6 Research Unit in Molecular Pathogenesis and Immunology of Infectious Diseases, Thammasat University, Pathumthani, Thailand

* upornpim@tu.ac.th, p.akswn@gmail.com

**Data Availability Statement:** All relevant data are within the manuscript and its Supporting Information file S1.

## Abstract

Coronavirus disease 2019 (COVID-19), a highly contagious pathogenic viral infection caused by severe acute respiratory syndrome coronavirus 2 (SARS-CoV-2) has spread rapidly and remains a challenge to global public health. COVID-19 patients manifest various symptoms from mild to severe cases with poor clinical outcomes. Prognostic values of novel markers, including neutrophil-to-lymphocyte ratio (NLR), platelet-to-lymphocyte ratio (PLR) and C-reactive protein to lymphocyte ratio (CLR) calculated from routine laboratory parameters have recently been reported to predict severe cases; however, whether this investigation can guide oxygen therapy in COVID-19 patients remains unclear. In this study, we assessed the ability of these markers in screening and predicting types of oxygen therapy in COVID-19 patients. The retrospective data of 474 COVID-19 patients were categorized into mild and severe cases and grouped according to the types of oxygen therapy requirement, including noninvasive oxygen support, high-flow nasal cannula and invasive mechanical ventilator. Among the novel markers, the ROC curve analysis indicated a screening cutoff of $CRP \geq 30.0$ mg/L, $NLR \geq 3.0$ and $CLR \geq 25$ in predicting the requirement of any type of oxygen support. The NLR and CLR with increasing cut-off values have discriminative power with high accuracy and specificity for more effective oxygen therapy with a high-flow nasal cannula ($NLR \geq 6.0$ and $CLR \geq 60$) and mechanical ventilator ($NLR \geq 8.0$ and $CLR \geq 80$). Our study thus identifies potential markers to differentiate the suitable management of oxygen therapy in COVID-19 patients at an earlier time for improving disease outcomes with limited respiratory support resources.

## Introduction

Coronavirus disease 2019 (COVID-19) is a highly contagious viral infectious disease caused by severe acute respiratory syndrome coronavirus 2 (SARS-CoV-2), which first emerged in

**Funding:** This study was supported by the Thammasat University Research Unit in Molecular Pathogenesis and Immunology of Infectious Diseases. The funders had no role in study design, data collection and analysis, decision to publish, or preparation of the manuscript.

**Competing interests:** The authors have declared that no competing interests exist.

Wuhan, China in December 2019, before causing a global pandemic [1]. The clinical outcomes of COVID-19 can be categorized as asymptomatic, mild, moderate or severe diseases [2]. Patients with poor outcomes showed a series of respiratory symptoms such as pneumonia and acute respiratory distress syndrome (ARDS) associated with a high mortality rate. Early prognosis is crucial for identifying potentially severe or critical cases, and for timely treatment [3].

The unregulated release of pro-inflammatory cytokines was suggested to cause lung damage in patients with severe COVID-19 [4]. The pattern of hyperinflammatory immune responses can determine a poor clinical outcome; particularly, blood parameters of inflammatory markers can help identifying disease severity early [5]. Several studies reported that biomarkers in routine laboratory hematological, biochemical and immunological tests were associated with disease severity [6–8]. More than 80% of severe COVID-19 cases had lymphopenia and thrombocytopenia [9]. Coagulogram and D-dimer were related to the development of disseminated intravascular coagulation (DIC) in severe and critical patients [10]. Liver enzymes and cardiac markers were used to monitor COVID-19 patients with multi-organ failure and myocardial injury [8]. Moreover, enhanced levels of inflammatory markers such as C-reactive protein (CRP) and IL-6 were significantly associated with disease severity [8, 11]. The related studies indicated that patients with a high level of CRP (more than 30 mg/dL) were suggested for oxygen therapy with a mechanical ventilator and steroid drug use [6, 7]. A more recent study indicated that serial CRP, neutrophil, and lymphocyte counts during the first three days of hospitalization can estimate supplemental oxygen requirement in patients with COVID-19 [12].

Recently, several emerging biomarkers such as neutrophil to lymphocyte ratio (NLR) [1, 13], platelet to lymphocyte ratio (PLR) [14] and C-reactive protein to lymphocyte ratio (CLR) [15] were used to assess COVID-19 progression and were more specific in the early triage of admission than was routine laboratory testing [16]. Many patients with mild symptoms can potentially progress to severe or critical stages and have different management with oxygen support [3]; therefore, a simple and efficient predictor is vital to provide increased attention care to reduce the mortality of COVID-19 patients. Our study aimed to determine whether these novel biomarkers could be useful in assessing the oxygen therapy requirement in COVID-19 patients to prevent a high risk of developing respiratory failure and severe diseases. This study might support risk assessment and the appropriate application of therapeutic strategies.

## Materials and methods

### Study design and participants

This retrospective cohort single-center study was conducted in confirmed COVID-19 patients at Saraburi Regional Hospital in Thailand during the third wave (delta variant outbreak) from 1 May 2021 to 31 August 2021. We included all adults with age $\geq$ 18 years old who were detected with SARS-CoV-2 by qRT-PCR on a nasopharyngeal swab. All patients had a routine laboratory examination, including complete blood count and C-reactive protein at the first time of admission. Patients with an illness duration of more than seven days and with a medical history or treatment that altered their blood parameters and CRP such as active cancer, chemotherapy, corticosteroid therapy, infection or hematological malignancies were excluded.

### Data collection

The data of enrolled cases, including demographic information, underlying disease, medical history, clinical outcome, oxygen therapy and laboratory findings, were retrospectively collected from the patient's electronic medical records. According to the COVID-19 clinical

practice guidelines of the medical services department of Thailand, severe cases were mainly associated with acute respiratory distress syndrome and an indication for using invasive oxygen support. Mild/moderate cases were defined by improved clinical signs and requirements for simple oxygen therapy. The oxygen therapy was classified as noninvasive oxygen support (oxygen mask bag and low flow oxygen cannula), high-flow nasal cannula and invasive mechanical ventilator, respectively. The routine laboratory data in the primary endpoint within seven days before admission consisted of a complete blood count and C-reactive protein. The CRP was measured by a nephelometry assay using MISPA-i3 Protein Analyzer (MEDIMIND Inc., Himachal Pradesh, India). The novel inflammatory markers, including neutrophil to lymphocyte ratio (NLR), platelet to lymphocyte ratio (PLR) and C-reactive protein to lymphocyte ratio (CLR) were calculated from a hemogram using an XN-3000 Automated Hematology Analyzer (Sysmex Co., Thailand).

## Ethics

This retrospective cohort study was approved by the institutional ethics committee of Saraburi Hospital (SRBR64-035, EC031/2564) and Thammasat University (ECSc128/2564) in Thailand. The requirement for informed consent was waived due to minimum risk to the patient and no identifiable information.

## Statistical analysis

This study was evaluated by the IBM SPSS Statistics version 25.0 statistical package program (Chicago, IL, USA) and GraphPad Prism software version 9.0 (San Diego, CA, USA). The continuous data were analyzed by providing the number of units (n), percentage (%) and median. The compliance of the categorical data and the continuous covariate was performed using independent t-tests and the Mann-Whitney U test, respectively. A one-way analysis of variance (ANOVA) was used to compare the differences in the median of multiple groups. The assessment of a threshold to discriminate between severe and mild cases and oxygen support requirements was performed by receiving operating characteristics (ROC) curves. The cutoff point was determined and chosen by Youden's index based on the appropriate sensitivity and specificity for each oxygen mechanism interventions. $P < 0.05$ value was considered statistically significant.

## Results

### Clinical characteristics of COVID-19 patients and types of oxygen support

A total of 474 patients with COVID-19 infection were included in this analysis. Of them, 243 were classified as mild cases (51.3%) and 231 as severe cases (48.7%). Their clinical severity was then correlated with their characteristics, including age, sex, body mass index (BMI) and comorbidity patterns (Table 1). The median age of mild (54 years; 19–87 years) and severe cases (60 years; 22–92) differed significantly. Moreover, we did not observe a difference in the levels of BMI associated with disease severity. For all cases, the most common comorbidities were two or more non-communicable diseases (hypertension, diabetes, gout, obesity, etc.; n = 123, 25.9%), only hypertension (n = 67, 14.1%), only diabetes (n = 47, 9.9%) and chronic kidney disease (n = 31, 6.6%). Patients with comorbidity conditions of diabetes and chronic kidney disease but not hypertension were significantly associated with a severe disease outcome.

All patients were evaluated for the length of hospital stay as indicated by the total number of days they spent in the hospital from the date of admission to the date of discharge or death. The median length of hospital stay among severe cases was found to exceed 14 days, while that

**Table 1. Clinical characteristics of mild and severe COVID-19 patients.**

| Characteristics | Total | Mild cases | Severe cases | P-value |
|---|---|---|---|---|
| Number of cases (%) | 474 | 243 (51.3%) | 231 (48.7%) | |
| Age (Median; Min–Max) | | 54 (19–87) | 60 (22–92) | ≤ 0.001 |
| < 60 years | 254 (53.6%) | 148 (58.3%) | 106 (41.7%) | ≤ 0.001 |
| ≥ 60 years | 220 (46.4%) | 95 (43.2%) | 125 (56.8%) | ≤ 0.001 |
| Sex | | | | |
| Male | 198 (41.8%) | 96 (48.5%) | 102 (51.5%) | 0.305 |
| Female | 276 (58.2%) | 147 (53.3%) | 129 (46.7%) | 0.305 |
| Body Mass Index (BMI) | | 26.5±6.6 | 27.4±7.2 | 0.174 |
| < 25.0 | 180 (38.0%) | 98 (54.4%) | 82 (45.6%) | 0.674 |
| ≥ 25.0 | 223 (62.0%) | 122 (52.4%) | 111 (47.6%) | 0.674 |
| Comorbidity | | | | |
| No known disease | 133 (28.1%) | 100 (75.2%) | 33 (24.8%) | ≤ 0.001 |
| Diabetes | 47 (9.9%) | 17 (36.2%) | 30 (63.8%) | 0.029 |
| Hypertension | 67 (14.1%) | 33 (49.3%) | 34 (50.7%) | 0.722 |
| Non-communicable diseases (two or more NCDs) | 123 (25.9%) | 49 (39.3%) | 74 (60.2%) | 0.003 |
| Chronic kidney disease (CKD) | 31 (6.6%) | 9 (29.0%) | 22 (71.0%) | 0.010 |
| Cerebrovascular accident (CVA) | 16 (3.4%) | 8 (50.0%) | 8 (50.0%) | 0.918 |
| Other (e.g., asthma, cancer, SLE, etc.) | 57 (12.0%) | 27 (47.4%) | 30 (52.6%) | 0.530 |

among mild disease was shorter than 7 days (Fig 1A). All COVID-19 patients with mild cases fully recovered, while 39% of severe diseases (90 out of 231 cases) were associated with mortality (Fig 1B). Among mild cases, 33.3% required no oxygen support, while 66.7% needed noninvasive oxygen support (Fig 1C). All severe COVID-19 patients required oxygen support as 61.9% of the severe cases required high-flow nasal cannula, and 31.6% received a mechanical ventilator (Fig 1D).

## Routine laboratory findings and novel biomarkers in COVID-19 patients

The laboratory parameters including complete blood count and C-reactive proteins of all patients on admission were evaluated and the levels of these values were compared between mild and severe diseases (Table 2). Compared with mild cases, we found no significant difference in the red blood cell count, hemoglobin, hematocrit, monocyte and platelet count in severe cases. Indeed, the median levels of white blood cell count, neutrophils and C-reactive proteins were significantly greater in patients with severe cases than in mild cases. Whereas, the median levels of absolute lymphocytes in COVID-19 patients with severe cases were markedly less than those with mild ones. The median concentrations of CRP in severe COVID-19 patients (80.9; 0.5–275.2 mg/L) were significantly higher than those in the mild cases (39.8; 0.5–221.2 mg/L).

Besides these routine laboratory parameters, the related evidence indicated that the emerging laboratory parameters such as neutrophil-lymphocyte ratio (NLR), platelet-lymphocyte ratio (PLR) and a recently described prognostic marker C-reactive protein (CRP)/lymphocyte ratio (CLR) have been considered useful indicators for diagnosis and prognosis of various infectious and inflammatory diseases, including cancers, autoimmune disease as well as COVID-19. We calculated these values for each of the patients. The median values of these parameters in patients with severe disease were significantly higher than in those with mild disease (Table 3), indicating the possibility of predictive biomarkers for evaluating the severity and guiding therapy in COVID-19 patients.

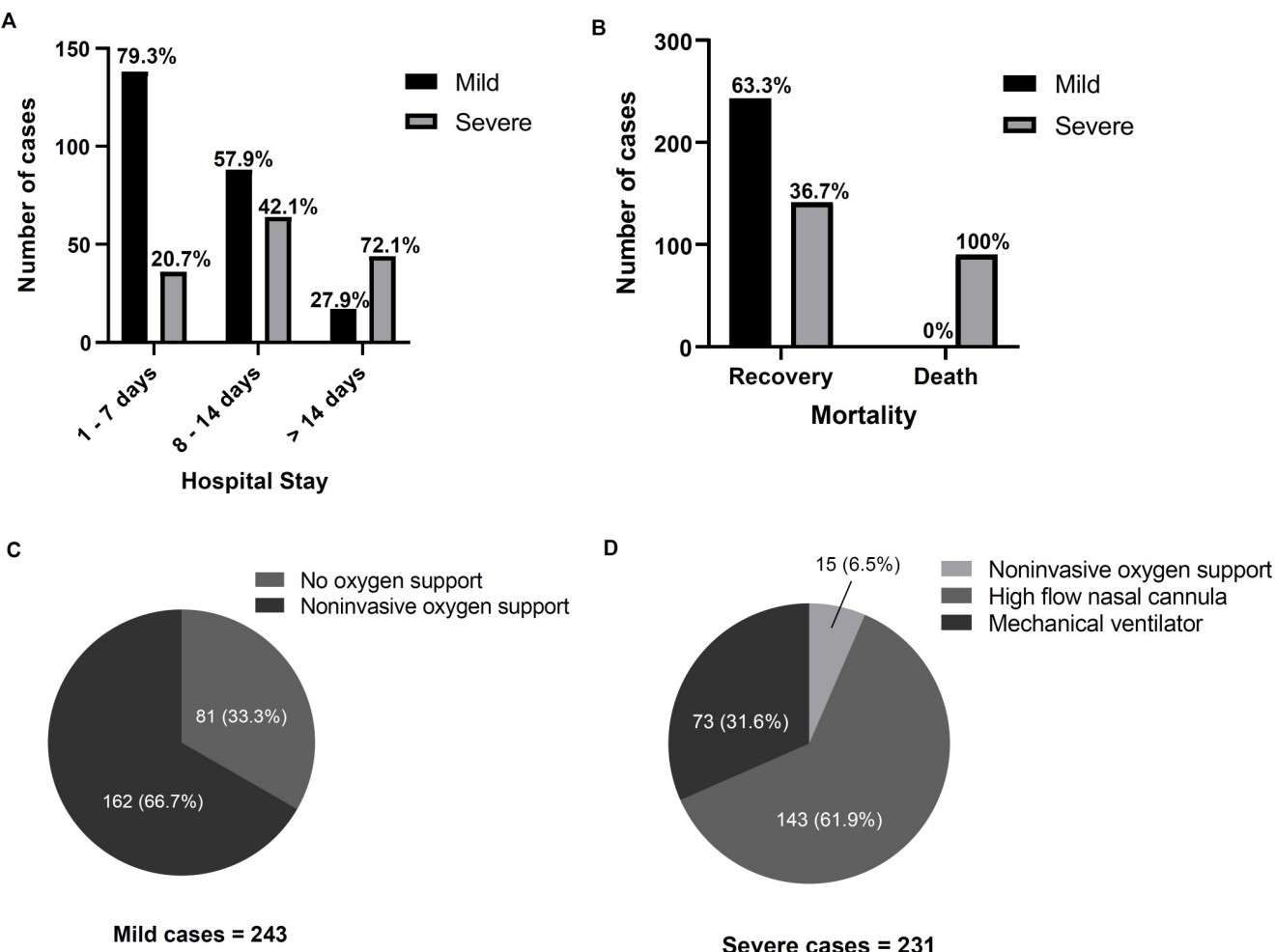

**Fig 1. Mortality and oxygen support requirements in mild and severe COVID-19 infection. (A)** Comparison of the length of hospital stay for mild and severe cases. The hospital stay of this study was divided into three groups: 1–7 days, 8–14 days and more than 14 days. **(B)** The mortality was compared, i.e., recovery and death in mild and severe COVID-19 infections. **(C and D)** The pies of oxygen support types used in these patients. The significance was determined using Mann-Whitney U test * P ≤ 0.05, ** P ≤ 0.01 *** and P ≤ 0.001 (compared with data in mild cases).

**Table 2. Routine laboratory findings of mild and severe COVID-19 patients.**

| Hematologic and Inflammatory parameters | Mild cases | Severe cases | P-value |
|---|---|---|---|
| WBC count ($10^3$ cell/μL) | 6.4 (2.1–19.5) | 8.3 (2.0–23.7) | ≤ 0.001 |
| RBC count ($10^6$ cell/μL) | 4.71±0.82 | 4.67±0.87 | 0.589 |
| Hemoglobin (g/dL) | 12.8±2.0 | 12.6±2.4 | 0.230 |
| Hematocrit (%) | 38.0 (11.0–55.0) | 37.6 (15.0–52.0) | 0.346 |
| Absolute neutrophil ($10^3$ cell/μL) | 4.39 (1.00–18.20) | 6.72 (1.00–21.52) | ≤ 0.001 |
| Absolute lymphocyte ($10^3$ cell/μL) | 1.24 (0.00–3.71) | 0.93 (0.00–2.43) | ≤ 0.001 |
| Absolute monocyte ($10^3$ cell/μL) | 0.36 (0.00–3.00) | 0.34 (0.00–2.00) | 0.292 |
| Platelet count ($10^3$ cell/μL) | 229 (36–690) | 222 (18–728) | 0.391 |
| C-reactive protein (mg/L) | 39.8 (0.5–221.2) | 80.9 (0.5–275.2) | ≤ 0.001 |

Abbreviations: WBC, white blood cell; RBC, red blood cell.

**Table 3. The novel inflammatory marker of mild and severe cases with COVID-19.**

| Novel inflammatory markers | Mild cases | Severe cases | P-value |
|---|---|---|---|
| NLR | 3.6 (0.4–31.5) | 7.0 (0.7–32.0) | ≤ 0.001 |
| PLR | 185.5 (27.6–780.7) | 240.3 (42.3–931.0) | ≤ 0.001 |
| CLR | 36.0 (0.2–272.2) | 76.2 (0.2–422.3) | ≤ 0.001 |

Abbreviations: NLR, neutrophil/lymphocyte ratio; PLR, platelet/lymphocyte ratio; CLR, C-reactive protein/lymphocyte ratio.

## Association between the novel markers and oxygen support requirements in COVID-19 patients

Because total leukocyte count, neutrophils, lymphocytes, CRP, NLR, PLR and CLR were markedly altered in patients with severe disease, we then further evaluated the possibility of using these markers in guiding types of oxygen therapy in COVID-19 patients by analyzing the median values among patients receiving oxygen mask bag, low flow oxygen cannula, high-flow nasal cannula and invasive mechanical ventilator. Compared with COVID-19 patients not requiring oxygen support, the novel markers were significantly higher in COVID-19 patients who need oxygen therapy of any types (Table 4 and Fig 2). Among these novel markers, the differentially enhanced values of NLR, PLR and CLR were observed in patients receiving high-flow nasal cannula (NLR: 6.1; 0.7–32.0, PLR: 250.0; 42.3–931.0 and CLR: 71.3; 0.2–382.7) compared to those receiving noninvasive oxygen support (NLR: 4.4; 0.4–31.5, PLR: 194.2; 27.6–780.7 and CLR: 51.3; 0.7–272.2). However, significantly increased levels of NLR and CLR were observed in patients requiring invasive mechanical ventilator (NLR: 8.3; 2.0–31.4 and CLR: 87.0; 2.1–422.3) compared to those receiving noninvasive oxygen support (NLR: 4.4; 0.4–31.5 and CLR: 51.3; 0.7–272.2). These data indicate that the novel markers, particularly NLR and CLR could be potentially used to predict more effective oxygen therapy need (Table 4 and Fig 2).

To assess the effectiveness of these markers in screening and predicting types of oxygen support management in COVID-19 patients, the ROC curve analysis was performed and the difference in the area under the curve (AUC) was tested between patients not requiring oxygen and requiring any type of oxygen support. The area under the ROC curve (AUC) indicated a screening power of CRP ≥ 30.0 mg/L (AUC = 0.74, 95% CI; 0.679–0.801, P ≤ 0.001), exhibited sensitivity of 75.6% and specificity of 59.3%, NLR ≥ 3.0 (AUC = 0.74, 95% CI; 0.682–0.797, P ≤ 0.001), exhibited sensitivity of 76.8% and specificity of 59.3% and CLR ≥ 25 (AUC = 0.764, 95% CI; 0.705–0.824, P ≤ 0.001), exhibited sensitivity of 75.8% and specificity of 60.5% (Fig 3 and Table 5).

**Table 4. The inflammatory marker among COVID patients requiring different types of oxygen support.**

| Parameters | No O$_2$ support | Noninvasive oxygen support | High-flow nasal cannula | Mechanical ventilator |
|---|---|---|---|---|
| Total WBC | 6.0 (2.1–15.2) | 6.8 (2.5–19.5) | 7.7 (2.3–23.0) | 9.0 (2.0–23.7) |
| Neutrophil | 3.70 (1.3–15.21) | 4.89 (0.8–18.20) | 6.11 (1.43–19.28) | 7.48 (1.48–21.52) |
| Lymphocyte | 1.55 (0.23–3.41) | 1.09 (0.34–3.71) | 0.97 (0.18–2.43) | 0.85 (0.12–2.20) |
| CRP | 20.4 (0.5–167.6) | 56.3 (0.5–221.2) | 71.3 (0.5–242.6) | 91.9 (2.1–275.2) |
| NLR | 2.5 (0.5–17.9) | 4.4 (0.4–31.5) | 6.1 (0.7–32.0) | 8.3 (2.0–31.4) |
| PLR | 158.6 (28.4–680.0) | 194.2 (27.6–780.7) | 250.0 (42.3–931.0) | 239.6 (68.7–888.2) |
| CLR | 15.7 (0.2–122.9) | 51.3 (0.7–272.2) | 71.3 (0.2–382.7) | 87.0 (2.1–422.3) |

Abbreviations: CRP, C-reactive protein; NLR, neutrophil/lymphocyte ratio; PLR, platelet/lymphocyte ratio; CLR, C-reactive protein/lymphocyte ratio.

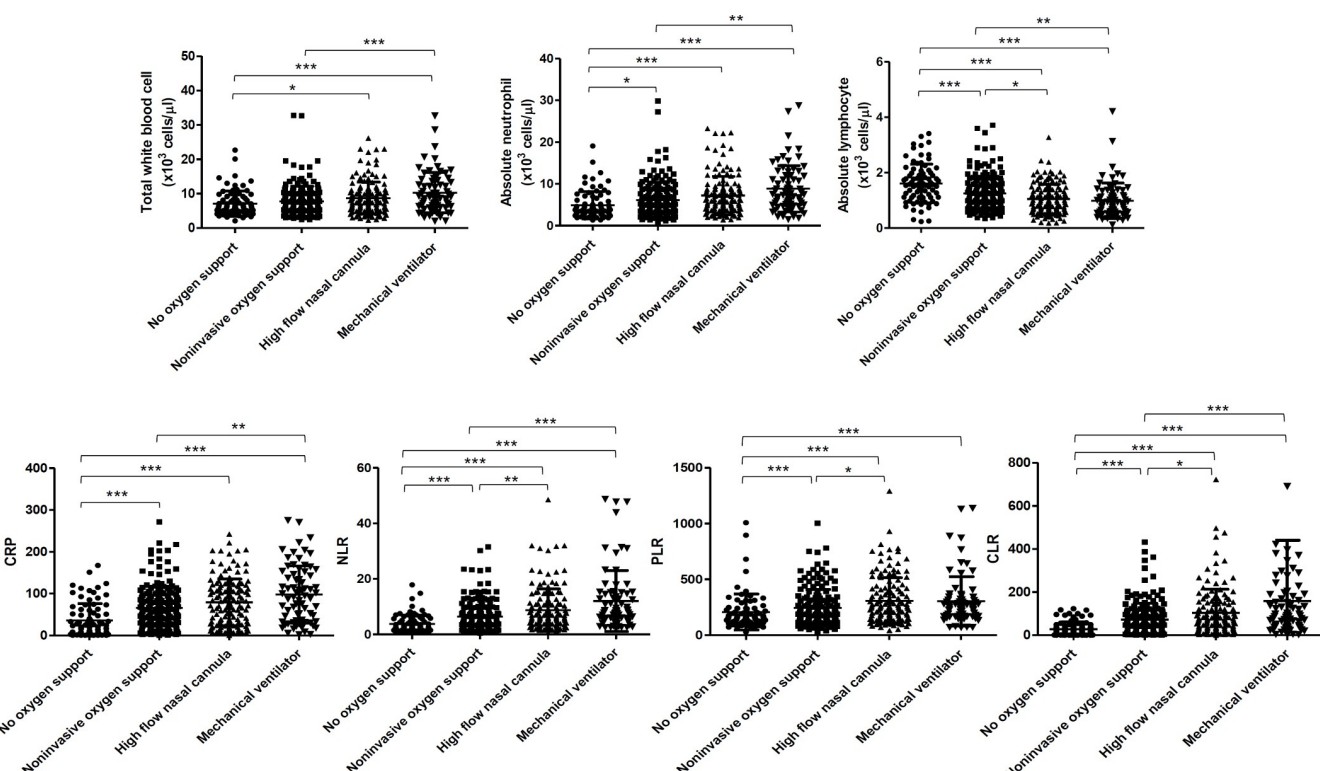

**Fig 2. Association between potential parameters and types of oxygen support among COVID-19 patients.** The illustration of total white blood cell count, absolute neutrophil count, absolute lymphocyte count, C-reactive protein (CRP), neutrophil/lymphocyte ratio (NLR), platelet/lymphocyte ratio (PLR) and C-reactive protein/lymphocyte ratio (CLR) in association with types of oxygen support. The requirement of oxygen support in this study was divided into four groups: no oxygen support, noninvasive oxygen support (oxygen mask bag and low flow oxygen cannula), high-flow nasal cannula and mechanical ventilator, respectively. Error bars denoted mean ± SD. The significance was determined by one-way ANOVA and Dunn's Multiple Comparison Test * P ≤ 0.05, ** P ≤ 0.01 and *** P ≤ 0.001, respectively.

To evaluate the potential utility of these markers in differentiating the requirement for more effective oxygen therapy, we further analyzed the discriminative capacity by increasing the cut-off value. With this determination, a cut-off NLR value of ≥ 6.0 and CLR value of ≥ 60 showed better prognostic ability and high specificity among the inflammatory markers in determining the requirement of high-flow nasal cannula (NLR: AUC = 0.777, 95% CI; 0.714–0.84, P ≤ 0.001 and CLR: AUC = 0.788, 95% CI; 0.725–0.851, P ≤ 0.001), with the sensitivity of 52.4% and specificity of 80.2% for NLR and the sensitivity of 57.3% and specificity of 82.7% for CLR. A cut-off value of NLR ≥ 8.0 and CLR ≥ 80 well-defined the requirement of the mechanical ventilator with high accuracy (NLR: AUC = 0.856, 95% CI; 0.799–0.913, P ≤ 0.001 and CLR: AUC = 0.826, 95% CI; 0.762–0.891, P ≤ 0.001) and diagnostic values (NLR: sensitivity 50.7% and specificity 92.6% and CLR: sensitivity 53.4% and specificity 85.2) (Fig 3 and Table 5).

To investigate the potential risk factors for the need for oxygen therapy in COVID-19 patients, univariate analyses were performed. As illustrated in Table 5, the analysis demonstrated that the CRP, NLR and CLR were associated with an odds ratio (OR) of 4.500, 4.827 and 4.672 for oxygen support requirement, respectively. Particularly, the NLR and CLR showed even higher odds ratio when associated with the requirement of more effective oxygen therapy with a high-flow nasal cannula (NLR: odds ratio: 4.481, 95% CI: 2.368–8.480 and CLR: odds ratio: 6.433, 95% CI: 3.310–12.503) or with a mechanical ventilator (NLR: odds ratio: 12.847, 95% CI: 4.970–33.210 and CLR: odds ratio: 6.596, 95% CI: 3.065–14.193) (Table 5).

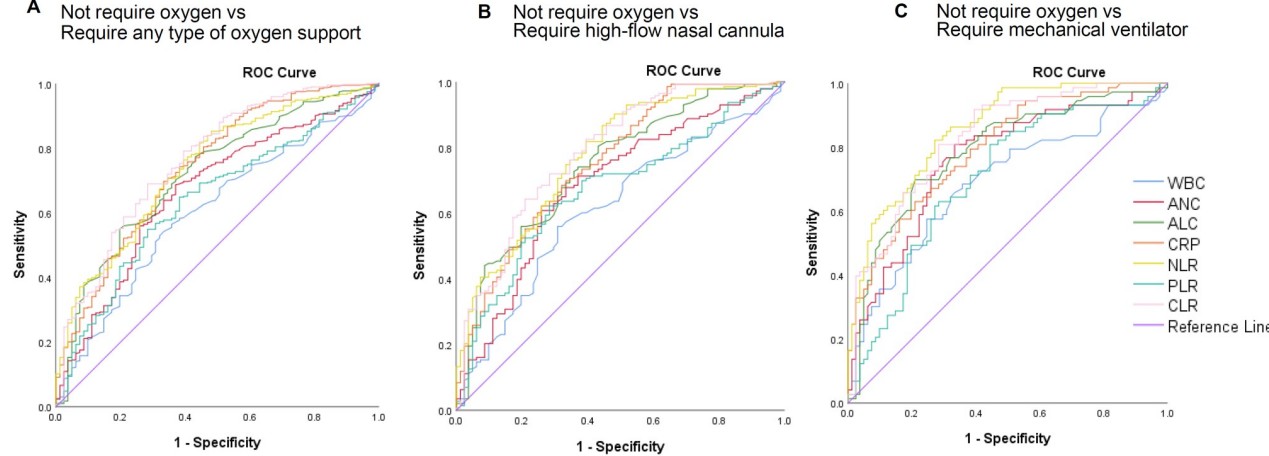

**Fig 3. Receiver operating characteristic (ROC) curve analysis of routine laboratory and novel markers for oxygen support requirements. (A)** The AUC of white blood cell count (WBC), absolute neutrophil count (ANC), absolute lymphocyte count (ALC), C-reactive protein (CRP), neutrophil/lymphocyte ratio (NLR), platelet/lymphocyte ratio (PLR) and C-reactive protein/lymphocyte ratio (CLR) for requiring any type of oxygen support was 0.603, 0.666, 0.719, 0.740, 0.740, 0.638 and 0.764 respectively. The ROC analysis showed the highest AUC value in CLR for the requirement of any type of oxygen. **(B)** The AUC of WBC, ANC, ALC, CRP, NLR, PLR and CLR for requiring oxygen support with high-flow nasal cannula was 0.616, 0.686, 0.744, 0.756, 0.777, 0.672 and 0.788, respectively. The CLR had the greatest AUCs for guiding oxygen therapy with a high-flow nasal cannula. **(C)** The AUC of WBC, ANC, ALC, CRP, NLR, PLR and CLR for need of mechanical ventilator was 0.688, 0.760, 0.785, 0.794, 0.856, 0.699 and 0.826, respectively. The ROC analysis showed the highest AUC value in NLR and CLR for guiding mechanical ventilator treatment.

**Table 5. Assessment of the inflammatory markers in predicting types of oxygen support management in COVID-19 patients.**

| Markers | AUC (95% CI) | Cut-off | %SEN | %SPE | %PPV | %NPV | OR (95% CI) | P-value |
|---|---|---|---|---|---|---|---|---|
| Not required oxygen vs Required any type of oxygen support | | | | | | | | |
| CRP | 0.740 (0.679–0.801) | 30.0 | 75.6 | 59.3 | 90.0 | 33.3 | 4.500 (2.731–7.415) | ≤ 0.001 |
| NLR | 0.740 (0.682–0.797) | 3.0 | 76.8 | 59.3 | 90.1 | 34.5 | 4.827 (2.924–7.970) | ≤ 0.001 |
| CLR | 0.764 (0.705–0.824) | 25.0 | 75.8 | 60.5 | 90.3 | 34.0 | 4.803 (2.908–7.933) | ≤ 0.001 |
| Not required oxygen vs Required high-flow nasal cannula | | | | | | | | |
| CRP | 0.756 (0.690–0.822) | 70.0 | 51.7 | 80.2 | 82.2 | 48.5 | 4.357 (2.302–8.245) | ≤ 0.001 |
| NLR | 0.777 (0.714–0.840) | 6.0 | 52.4 | 80.2 | 82.4 | 48.9 | 4.481 (2.368–8.480) | ≤ 0.001 |
| CLR | 0.788 (0.725–0.851) | 60.0 | 57.3 | 82.7 | 85.4 | 52.3 | 6.433 (3.310–12.503) | ≤ 0.001 |
| Not required oxygen support vs Required mechanical ventilator | | | | | | | | |
| CRP | 0.794 (0.725–0.863) | 90.0 | 50.7 | 86.4 | 77.1 | 66.0 | 6.540 (2.986–14.325) | ≤ 0.001 |
| NLR | 0.856 (0.799–0.913) | 8.0 | 50.7 | 92.6 | 86.0 | 67.6 | 12.847 (4.970–33.210) | ≤ 0.001 |
| CLR | 0.826 (0.792–0.891) | 80.0 | 53.4 | 85.2 | 76.5 | 67.0 | 6.596 (3.065–14.193) | ≤ 0.001 |

Abbreviations: CRP, C-reactive protein; NLR, neutrophil/lymphocyte ratio; PLR, platelet/lymphocyte ratio; CLR, C-reactive protein/lymphocyte ratio.

## Discussion

The severe acute respiratory syndrome-coronavirus 2 (SARS-CoV-2) causes COVID-19 disease ranging from mild symptoms to pulmonary disease with acute respiratory distress syndrome and multiple organ dysfunction [17]. An overactive immune response has been associated with a severe COVID-19 disease outcome [2, 4]. Several biomarkers, including complete blood count, coagulogram, ferritin and inflammatory markers such as C-reactive protein (CRP) and IL-6 are currently used for predicting disease severity, hospitalization and mortality [6–8]. More recently, novel inflammatory markers such as the neutrophil/lymphocyte ratio (NLR), platelet/lymphocyte ratio (PLR) and C-reactive protein (CRP)/lymphocyte ratio (CLR) have been recognized as potential biomarkers for the diagnosis and prognosis of COVID-19 infection [16, 18, 19]. In this study, we further demonstrated that these markers can differentiate the disease severity of COVID-19 patients and predict the types of oxygen therapy.

Among the routine hematological and inflammatory parameters, we indeed observed significant changes in the level of neutrophils, CRP and lymphocytes in severe COVID-19 patients. The enhanced levels of neutrophil and CRP were associated with the induction of inflammatory cytokines, resulting in severe symptoms in patients with COVID-19 [3, 20]. Thus, these markers were useful in assessing patient outcome together with clinical manifestation. Similar to other studies showing lymphocytopenia in hospitalized patients, our study revealed a significantly lower absolute lymphocyte count in patients with severe disease. Not surprisingly, the ratio of neutrophil, CRP and lymphocytes indicated as NLR (7.0: 0.7–32.0) and CLR (76.2: 0.2–422.3) demonstrated markedly changed in severe cases. A related study indicated that an NLR greater than 6.5 may reflect the progression of the disease towards an unfavorable clinical outcome [21]. A recent study showed that patients with NLR ≥ 2.6937 were at high risk with the potential to develop worsened and serious clinical outcomes [22]. In previous study, the cutoff value for CLR ≥ 78.3 can be regarded as patients with severe disease, and ≥ 159.5 can predict mortality [15]. Together with our study, the NLR and CLR showed the most consistent results across studies in predicting COVID-19 disease severity.

Because of limited oxygen therapy resources, predicting COVID-19 patients who require close monitoring, including supplementary oxygen either noninvasive ventilation or high-flow nasal oxygen or invasive mechanical ventilation is important [23, 24]. Recent studies indicate the prediction model that can identify patients at high risk of respiratory failure at an early stage using the CRP, hypertension, age, neutrophil and lymphocyte (CHANeL) predictors [12]. They showed that high CRP and neutrophil counts and low lymphocyte counts were associated with a requirement for supplementary oxygen support and a worse outcome [12]. We further evaluated key routine laboratory and inflammatory parameters for the need for different types of oxygen support. The NLR and CLR for screening respiratory failure with the cut-off values of ≥ 3.0 and ≥ 25.0 were the markers most useful in predicting the requirement of oxygen therapy. Then, the cut-off values of ≥ 6.0 of NLR and ≥ 60.0 of CLR were able to further determine patients with severe pneumonia management using a high-flow nasal cannula. In some COVID-19 patients, the dyspnea rapidly steps up from severe to critically ill. The cut-off values of ≥ 8.0 and ≥ 80.0 for NLR and CLR can be utilized for a highly specific patient requirement for high effectiveness of oxygen support, especially mechanical ventilation. In real-world resources, the capacity and access to the high-flow nasal cannula or invasive mechanical ventilator is limited. Therefore, choosing the cut-off value step by step can immediately help and differentiate suitable management. Previous studies indicated that the maximal IL-6 level before intubation showed the strongest association with the need for mechanical ventilation, followed by maximal CRP level in COVID-19 patients, with the optimal cutoff

value of IL-6 level > 80 pg/mL and of CRP level > 97 mg/L during the course of the disease from the evaluation cohort [25]. The increased neutrophils count was also recognized as a good predictor of the requirement of the invasive or non-invasive mechanical ventilator [18].

Altogether, the results of our study demonstrated the NLR and CLR as key promising markers for the requirement of more effective oxygen support with a high-flow nasal cannula or invasive mechanical ventilator in COVID-19 patients, providing a better understanding of the consistency and magnitude of laboratory parameters for guiding the specific therapeutic interventions for improved disease outcomes. Our study has several notable limitations. First, the data were obtained from a single hospital center in Thailand. Second, it is retrospective in nature. Hence, the data are subject to other confounding factors regardless of the number of exclusion criteria added. Third, our study data were collected during the third wave of the COVID-19 pandemic in Thailand, when many patients with asymptomatic to mild cases were not admitted at the research hospital.

## Conclusion

The result of our study demonstrated the NLR and CLR as potentially reliable markers for predicting the requirement of supplementary oxygen therapy including high-flow oxygen nasal cannula and invasive mechanical ventilation in COVID-19 patients. We suggest that these emerging hematological and inflammatory markers may be useful in guiding the specific therapeutic management to improve the clinical outcome of COVID-19 patients with limited medical resources. However, a further study with a larger sample size in multiple centers is warranted.

## Supporting information

**S1 File. The original data of this study.**
(XLSX)

## Acknowledgments

We thank the Faculty of Allied Health Sciences and the Department of Medical Technology and Clinical Pathology and the Department of Medicine, Saraburi Hospital for their support.

## Author Contributions

**Conceptualization:** Peerapong Kamjai, Sivaporn Hemvimol, Narisa Kengtrong Bordeerat, Potjanee Srimanote, Pornpimon Angkasekwinai.

**Data curation:** Peerapong Kamjai.

**Formal analysis:** Peerapong Kamjai, Pornpimon Angkasekwinai.

**Funding acquisition:** Pornpimon Angkasekwinai.

**Investigation:** Peerapong Kamjai, Pornpimon Angkasekwinai.

**Methodology:** Peerapong Kamjai, Sivaporn Hemvimol, Narisa Kengtrong Bordeerat, Potjanee Srimanote, Pornpimon Angkasekwinai.

**Project administration:** Peerapong Kamjai, Pornpimon Angkasekwinai.

**Resources:** Sivaporn Hemvimol, Pornpimon Angkasekwinai.

**Software:** Pornpimon Angkasekwinai.

**Supervision:** Sivaporn Hemvimol, Narisa Kengtrong Bordeerat, Potjanee Srimanote, Pornpimon Angkasekwinai.

**Validation:** Peerapong Kamjai, Pornpimon Angkasekwinai.

**Visualization:** Peerapong Kamjai.

**Writing – original draft:** Peerapong Kamjai, Pornpimon Angkasekwinai.

**Writing – review & editing:** Peerapong Kamjai, Sivaporn Hemvimol, Narisa Kengtrong Bordeerat, Potjanee Srimanote, Pornpimon Angkasekwinai.

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
