## [Decision Letter · Decision Letter 0]

19 Sep 2022

PONE-D-22-17792Association between the novel inflammatory markers and oxygen support requirement in COVID-19 patientsPLOS ONE

Dear Dr. Angkasekwinai,

Thank you for submitting your manuscript to PLOS ONE. After careful consideration, we feel that it has merit but does not fully meet PLOS ONE’s publication criteria as it currently stands. Therefore, we invite you to submit a revised version of the manuscript that addresses the points raised during the review process.

We look forward to receiving your revised manuscript.

Kind regards,

Muhammad Tarek Abdel Ghafar, M.D

Academic Editor

PLOS ONE

Journal Requirements:

“This study was supported by the Thammasat University Research Unit in Molecular 329 Pathogenesis and Immunology of Infectious Diseases.”

“This study was supported by the Thammasat University Research Unit in Molecular Pathogenesis and Immunology of Infectious Diseases. The funders had no role in study design, data collection and analysis, decision to publish, or preparation of the manuscript.”

Reviewers' comments:

Reviewer's Responses to Questions

**Comments to the Author**

1. Is the manuscript technically sound, and do the data support the conclusions?

Reviewer #1: Yes

Reviewer #2: Yes

Reviewer #3: Yes

Reviewer #4: Yes

Reviewer #5: Yes

2. Has the statistical analysis been performed appropriately and rigorously? 

Reviewer #1: Yes

Reviewer #2: Yes

Reviewer #3: Yes

Reviewer #4: Yes

Reviewer #5: Yes

3. Have the authors made all data underlying the findings in their manuscript fully available?

Reviewer #1: Yes

Reviewer #2: Yes

Reviewer #3: Yes

Reviewer #4: Yes

Reviewer #5: Yes

4. Is the manuscript presented in an intelligible fashion and written in standard English?

Reviewer #1: Yes

Reviewer #2: Yes

Reviewer #3: Yes

Reviewer #4: Yes

Reviewer #5: No

5. Review Comments to the Author

Reviewer #1: This is a good manuscript. I read it with great interest. I acknowledge your effort in preparing this.

The only feedback I have is that of a minor revision.

I believe it needs a separate “conclusion/recommendation” segment which highlights the importance of your findings in light of their public and global health translations and implications.

Additionally, kindly make sure that your writing reconciles with PLOS guidelines.

Reviewer #2: This manuscript is technically sound and its conclusions are well supported by results. Articles of this topic are highly important in the fight to implement timely tools for better outcome of patients in need of the different oxygen support measures. Statistical analysis was performed appropriately and rigorously explained. Limitations were clearly described and no conflict of interest were declared.

Reviewer #3: Thanks to authors. This study on the Covid-19 pandemic which is a common problem of the whole world is very interesting. ın my opinion, this study will contribute to the current medical literature.

But, minor revision required.

Reviewer #4: Association between the novel inflammatory markers and oxygen support requirement in COVID-19 patients

Authors identified potential markers to differentiate suitable management of oxygen therapy in COVID-19 patients at earlier time for improving disease outcomes during limited resources of respiratory support.

Abstract: No comment

Introduction: No comment

Methods: No comment

Results: No comment

Discussion: No Comment

Conclusion: Non comment

This is a well designed, interesting study and the results are clear. These new markers seem to be helpful in the establishment of the prognosis of Covid-19 infection and the decision of the type of oxygen therapy. Markers are simple and easy to get routinely.

Reviewer #5: The manuscript by Kamjai and co-authors reported a retrospective, single-center study on ~470 COVID-19 patients during the delta variant outbreak in Thailand. Patients were divided into mild and severe categories based on oxygen supply requirements. The authors found that three markers, NLR, PLR, and CLR, significantly differ in mild and severe cases. ROC analysis revealed that NLR and CLR better predict oxygen requirements than other parameters.

The manuscript addresses an essential and timely question regarding medical resource triage, particularly for the local region, and will be of interest to the journal readers. The figures and tables are well constructed. Although the research appears sound, I have a few major concerns that prevent me from accepting the manuscript at the present stage.

1. The text needs significant editing for writing quality. The language of the manuscript is unclear and sometimes challenging to follow. Numerous grammar issues have been found. The authors should consider working with a copy editor to improve the readability of the text.

2. 'Novel markers' is a vague term. Consider rewriting the title to deliver the take-home message of the paper explicitly.

3. The abstract should be concise and report the key hypothesis, methods, results, and conclusions. It should not report numbers and statistics in detail. Line 41 is confusing.

4. It would be helpful to provide more details on the statistical analysis, particularly the ROC analysis and how cutoffs were determined.

6. PLOS authors have the option to publish the peer review history of their article (what does this mean?). If published, this will include your full peer review and any attached files.

Reviewer #1: No

Reviewer #2: No

Reviewer #3: No

Reviewer #4: No

Reviewer #5: No

---

## [Author Response · Author response to Decision Letter 0]

28 Sep 2022

Point-by-point Reply

Review Comments to the Author

Reviewer #1:

 This is a good manuscript. I read it with great interest. I acknowledge your effort in preparing this.

The only feedback I have is that of a minor revision.

I believe it needs a separate “conclusion/recommendation” segment which highlights the importance of your findings in light of their public and global health translations and implications.

Additionally, kindly make sure that your writing reconciles with PLOS guidelines.

Response: We thank the reviewer for this suggestion. In this revised manuscript, we have separated “conclusion” segment which highlights the significance of our findings as shown in line 323-330 in the current manuscript. We also recheck and ensure that the manuscript follows the PLOS guidelines.

Reviewer #2: 

This manuscript is technically sound and its conclusions are well supported by results. Articles of this topic are highly important in the fight to implement timely tools for better outcome of patients in need of the different oxygen support measures. Statistical analysis was performed appropriately and rigorously explained. Limitations were clearly described and no conflict of interest were declared.

Response: We thank the reviewer for the thoughtful review of our work and kind words.

Reviewer #3: 

Thanks to authors. This study on the Covid-19 pandemic which is a common problem of the whole world is very interesting. ın my opinion, this study will contribute to the current medical literature.

But, minor revision required.

Response: We appreciate the reviewer’s comments. We have carefully revised the current manuscript according to all reviewer’s comments.

Reviewer #4: 

Association between the novel inflammatory markers and oxygen support requirement in COVID-19 patients

Authors identified potential markers to differentiate suitable management of oxygen therapy in COVID-19 patients at earlier time for improving disease outcomes during limited resources of respiratory support.

Abstract: No comment

Introduction: No comment

Methods: No comment

Results: No comment

Discussion: No Comment

Conclusion: Non comment

This is a well-designed, interesting study and the results are clear. These new markers seem to be helpful in the establishment of the prognosis of Covid-19 infection and the decision of the type of oxygen therapy. Markers are simple and easy to get routinely.

Response: We thank the reviewer for taking time to review our work and kind comments.

Reviewer #5: 

The manuscript by Kamjai and co-authors reported a retrospective, single-center study on ~470 COVID-19 patients during the delta variant outbreak in Thailand. Patients were divided into mild and severe categories based on oxygen supply requirements. The authors found that three markers, NLR, PLR, and CLR, significantly differ in mild and severe cases. ROC analysis revealed that NLR and CLR better predict oxygen requirements than other parameters.

The manuscript addresses an essential and timely question regarding medical resource triage, particularly for the local region, and will be of interest to the journal readers. The figures and tables are well constructed. Although the research appears sound, I have a few major concerns that prevent me from accepting the manuscript at the present stage.

1. The text needs significant editing for writing quality. The language of the manuscript is unclear and sometimes challenging to follow. Numerous grammar issues have been found. The authors should consider working with a copy editor to improve the readability of the text.

Response: Because of the reviewer’s comments on the writing quality, the revised manuscript has been edited by the native English Editor through the Editing service. 

2. 'Novel markers' is a vague term. Consider rewriting the title to deliver the take-home message of the paper explicitly.

Response: We thank the reviewer for these suggestions. We revised the current manuscript title as “Evaluation of emerging inflammatory markers for predicting oxygen support requirement in COVID-19 patients”.

3. The abstract should be concise and report the key hypothesis, methods, results, and conclusions. It should not report numbers and statistics in detail. Line 41 is confusing.

Response: We appreciate suggestion by the reviewer. We have revised the abstract accordingly in the current manuscript as below:

 “Coronavirus disease 2019 (COVID-19), a highly contagious pathogenic viral infection caused by severe acute respiratory syndrome coronavirus 2 (SARS-CoV-2) has spread rapidly and remains a challenge to global public health. COVID-19 patients manifest various symptoms from mild to severe cases with poor clinical outcomes. Prognostic values of novel markers, including neutrophil-to-lymphocyte ratio (NLR), platelet-to-lymphocyte ratio (PLR) and C-reactive protein to lymphocyte ratio (CLR) calculated from routine laboratory parameters have recently been reported to predict severe cases; however, whether this investigation can guide oxygen therapy in COVID-19 patients remains unclear. In this study, we assessed the ability of these markers in screening and predicting types of oxygen therapy in COVID-19 patients. The retrospective data of 474 COVID-19 patients were categorized into mild and severe cases and grouped according to the types of oxygen therapy requirement, including noninvasive oxygen support, high-flow nasal cannula and invasive mechanical ventilator. Among the novel markers, the ROC curve analysis indicated a screening cutoff of CRP ≥ 30.0 mg/L, NLR ≥ 3.0 and CLR ≥ 25 in predicting the requirement of any type of oxygen support. The NLR and CLR with increasing cut-off values have discriminative power with high accuracy and specificity for more effective oxygen therapy with a high-flow nasal cannula (NLR ≥ 6.0 and CLR ≥ 60) and mechanical ventilator (NLR ≥ 8.0 and CLR ≥ 80). Our study thus identifies potential markers to differentiate the suitable management of oxygen therapy in COVID-19 patients at an earlier time for improving disease outcomes with limited respiratory support resources.”

4. It would be helpful to provide more details on the statistical analysis, particularly the ROC analysis and how cutoffs were determined.

Response: We appreciate helpful suggestion by the reviewer. We included more details on the ROC analysis and determination of cutoffs as “This study was evaluated by the IBM SPSS Statistics version 25.0 statistical package program (Chicago, IL, USA) and GraphPad Prism software version 9.0 (San Diego, CA, USA). The continuous data were analyzed by providing the number of units (n), percentage (%) and median. The compliance of the categorical data and the continuous covariate was performed using independent t-tests and the Mann-Whitney U test, respectively. A one-way analysis of variance (ANOVA) was used to compare the differences in the median of multiple groups. The assessment of a threshold to discriminate between severe and mild cases and oxygen support requirements was performed by receiving operating characteristics (ROC) curves. The cutoff point was determined and chosen by Youden’s index based on the appropriate sensitivity and specificity for each oxygen mechanism interventions. P < 0.05 value was considered statistically significant.” The additional information was included in the method session in line 109-119.

---

## [Decision Letter · Decision Letter 1]

11 Nov 2022

Evaluation of emerging inflammatory markers for predicting oxygen support requirement in COVID-19 patients

PONE-D-22-17792R1

Dear Dr. Angkasekwinai,

We’re pleased to inform you that your manuscript has been judged scientifically suitable for publication and will be formally accepted for publication once it meets all outstanding technical requirements.

Kind regards,

Muhammad Tarek Abdel Ghafar, M.D

Academic Editor

PLOS ONE

Additional Editor Comments (optional):

Reviewers' comments:

Reviewer's Responses to Questions

**Comments to the Author**

1. If the authors have adequately addressed your comments raised in a previous round of review and you feel that this manuscript is now acceptable for publication, you may indicate that here to bypass the “Comments to the Author” section, enter your conflict of interest statement in the “Confidential to Editor” section, and submit your "Accept" recommendation.

Reviewer #2: All comments have been addressed

Reviewer #4: All comments have been addressed

Reviewer #5: All comments have been addressed

2. Is the manuscript technically sound, and do the data support the conclusions?

Reviewer #2: Yes

Reviewer #4: Yes

Reviewer #5: Yes

3. Has the statistical analysis been performed appropriately and rigorously? 

Reviewer #2: Yes

Reviewer #4: Yes

Reviewer #5: Yes

4. Have the authors made all data underlying the findings in their manuscript fully available?

Reviewer #2: Yes

Reviewer #4: Yes

Reviewer #5: Yes

5. Is the manuscript presented in an intelligible fashion and written in standard English?

Reviewer #2: Yes

Reviewer #4: Yes

Reviewer #5: Yes

6. Review Comments to the Author

Reviewer #2: All reviewer's comments were properly addressed and writing quality improved by a native English editor service; therefore have no more comments, this paper is good to go from my point of view.

Reviewer #4: I have no more comments and concerns. The previous concerns were all addressed and included in the manuscript

Reviewer #5: Congratulations to the authors for a great job in the revision of this manuscript. The comments have been satisfactorily addressed and I recommend the acceptance of the manuscript.

7. PLOS authors have the option to publish the peer review history of their article (what does this mean?). If published, this will include your full peer review and any attached files.

Reviewer #2: No

Reviewer #4: No

Reviewer #5: No

---

## [Editor Report · Acceptance letter]

15 Nov 2022

PONE-D-22-17792R1 

Evaluation of emerging inflammatory markers for predicting oxygen support requirement in COVID-19 patients 

Dear Dr. Angkasekwinai:

I'm pleased to inform you that your manuscript has been deemed suitable for publication in PLOS ONE. Congratulations! Your manuscript is now with our production department. 

Kind regards, 

on behalf of

Prof Muhammad Tarek Abdel Ghafar 

Academic Editor

PLOS ONE